# Single Cell Center of Mass for the Analysis of BMP Receptor Heterodimers Distributions

**DOI:** 10.3390/jimaging7110219

**Published:** 2021-10-20

**Authors:** Hendrik Boog, Rebecca Medda, Elisabetta Ada Cavalcanti-Adam

**Affiliations:** 1Department of Cellular Biophysics-Growth Factor Mechanobiology, Max-Planck-Institute for Medical Research, 69120 Heidelberg, Germany; hendrik.boog@mr.mpg.de (H.B.); Rebecca.Medda@mr.mpg.de (R.M.); 2Institute for Pharmacy and Molecular Biotechnology (IPMB), Ruprecht-Karls-Universitaet Heidelberg, Im Neuenheimer Feld 364, 69120 Heidelberg, Germany

**Keywords:** receptor complex, distribution, center of mass, cell shape

## Abstract

At the plasma membrane, transmembrane receptors are at the interface between cells and their environment. They allow sensing and transduction of chemical and mechanical extracellular signals. The spatial distribution of receptors and the specific recruitment of receptor subunits to the cell membrane is crucial for the regulation of signaling and cell behavior. However, it is challenging to define what regulates such spatial patterns for receptor localization, as cell shapes are extremely diverse when cells are maintained in standard culture conditions. Bone morphogenetic protein receptors (BMPRs) are serine-threonine kinases, which build heteromeric complexes of BMPRI and II. These are especially interesting targets for receptor distribution studies, since the signaling pathways triggered by BMPR-complexes depends on their dimerization mode. They might exist as preformed complexes, or assemble upon binding of BMP, triggering cell signaling which leads to differentiation or migration. In this work we analyzed BMPR receptor distributions in single cells grown on micropatterns, which allow not only to control cell shape, but also the distribution of intracellular organelles and protein assemblies. We developed a script called ComRed (Center Of Mass Receptor Distribution), which uses center of mass calculations to analyze the shift and spread of receptor distributions according to the different cell shapes. ComRed was tested by simulating changes in experimental data showing that shift and spread of distributions can be reliably detected. Our ComRed-based analysis of BMPR-complexes indicates that receptor distribution depends on cell polarization. The absence of a coordinated internalization after addition of BMP suggests that a rapid and continual recycling of BMPRs might occur. Receptor complexes formation and localization in cells induced by BMP might yield insights into the local regulation of different signaling pathways.

## 1. Introduction

Bone morphogenetic proteins (BMPs) are growth factors involved in a plethora of cellular processes, like cell differentiation [1], adhesion, and migration [2,3]. Bone morphogenetic protein receptors (BMPRs) belong to the serine-threonine kinase family and become activated upon binding of BMP molecules to their extracellular domains, triggering downstream signaling [4]. BMPRs are present in cells either as a preformed complex (PFC) of BMPRI and BMPRII, or a BMP-induced signaling complex (BISC). The PFC is already assembled in absence of the growth factor, while in the BISC, BMP binding to BMPRI triggers the assembly of the complex [4]. The two complexes activate two different signaling cascades, a Smad-dependent pathway triggered by the PFC and a non-Smad pathway triggered by the BISC [4,5]. In both pathways, the complexes internalize with or without the cargo [6,7]. The PFC is internalized by clathrin-mediated endocytosis [4,8], while the BISC is uptaken via caveolae structures [5,9]. Although BMPRs were first identified in bone cells and cells of mesenchymal origin [10], these receptors are expressed in a variety of cells, including epithelial and endothelial cells [11,12]. In recent studies, the importance of BMPR-mediated signaling in endothelial cells was highlighted for the maintenance of tissue homeostasis [13,14,15]. Among the commonly used endothelial cell lines, human umbilical cord vein endothelial cells (HUVECs) express both BMPRI and II but have a low expression of the BMPR Ia subtype (also called Alk3), in favor of BMPRIb [12,16]. HUVECs respond to BMP-2, one member of the BMP growth factor family, by increasing their migratory behavior and angiogenesis. This response suggests that not only the specific expression of receptor subunits, but also the presence of ligand and receptor spatial organization might regulate adhesion-mediated responses triggered by BMPs [12,16]. Analysis of receptor localization can be categorized into three different approaches-the analysis of receptor interactions, velocities, and spatial distribution. The analysis of receptor interactions, looks at short range co-occurrences or binding events of receptors. Established methods are for example fluorescence resonance energy transfer (FRET) [17,18], colocalization [19,20,21], or fluorescence (cross-) correlation spectroscopy [22]. An example for velocity studies is fluorescence recovery after photobleaching (FRAP) [23].

Image correlation spectroscopy was used to analyze the receptor distributions of BMPRIa and BMPRII in COS7 and A431 cells. Here, a metric called cluster density, which is derived from image correlation spectroscopy, was used. BMPRIa formed clusters when cells were treated with BMP-2 for 2.5 h and when in absence of BMPRII. Moreover, a kinase inactive variant of BMPRII also led to aggregation of BMPRIa [24]. By using image correlation spectroscopy and FRET it was shown that BMPRI is shuffling towards BMPRII, which is located in precursors of clathrin-coated pits. Upon BMP induction, a significantly different distribution after inhibition of clathrin-mediated endocytosis could be observed [8]. However, these studies did not analyze spatial distributions within whole cells. A common problem of analyzing spatial distributions for whole cells are the nonuniform shapes and sizes of cells in standard culture conditions. Micropatterning techniques can restrict single cells to a uniform shape and size [25], which allows comparative analyses.

In the present study we developed the Center Of Mass Receptor Distribution (ComRed) script. ComRed analyzes the whole cell distributions of cellular receptors in fluorescence microscopy images, acquired after cell immunostaining. It uses data from upstream programs, for example Ilastik [26,27], that segment images automatically to compute the center of mass of receptor clusters. As a frame of reference, the center of mass of the nucleus was taken. The functionality of ComRed was tested with data simulations. Since receptor distributions of BMPRs were studied before [8,21,24], we also looked at BMPRs here using HUVECs, being an established primary cell model for the study of BMPRs [12,13,14,15,16,21]. Upon BMP treatment, HUVECs increase their motility and invasion as part of angiogenesis [12,15,16], and the spatial organization of BMPRs might play a substantial role in this process. Additionally, HUVECs express BMPRIb and BMPRII, while BMPRIa and Alk2 are only expressed slightly [12,16]. This simplifies the analysis of spatial distribution in HUVECs, as we expect no significant competition between BMPRIa and BMPRIb for complexing with BMPRII.

We compared receptor distributions in single cells that were grown on micropatterned substrates to control cell shape. We chose circle and crossbow shapes since cells on circular patterns adopt an unpolarized geometry due to isometric spreading, while cells on crossbow patterns adopt a polarized, migratory-like geometry [25,28]. Our results show that receptor distribution depends on cell polarization. Furthermore, we observed differences in BMPRIb and BMPRII distributions in unpolarized cells, indicating a preference for the BISC and the corresponding non-Smad pathways. ComRed is however not restricted to the analysis of BMPRs and in combination with micropatterning techniques it might be a useful tool to analyze the spatial distributions of any receptor of interest.

## 2. Results

In this study the spatial distribution of BMPR subunits Ib and II in fluorescence microscopy images was studied by using ComRed, a script which analyzes center of mass differences. A typical workflow to generate data using ComRed is illustrated in Figure 1. It consists of the following steps: 1. Coverslips are micropatterned to generate a pattern isolating single cells. In this study we used a circle and crossbow pattern to employ a polarized, migratory-like geometry in crossbow-shaped cells and an unpolarized geometry in circle-shaped cells. The micropatterned regions have a diameter of 50 µm to accommodate a single cell. 2. Cells are seeded on the micropatterned surfaces. After cell spreading, the micropatterned adhesive regions are completely occupied. In this study cells were subsequently exposed to BMP-2 added to the cell culture media for 15 min to trigger activation and internalization of the BMP-BMPRs complexes. Cells were fixed, immunostained, and imaged by confocal microscopy. 3. The images are segmented and the data are subsequently analyzed using ComRed, yielding comparisons of groups of images. ComRed is using center of mass calculations for its analysis, as described in Section 4. However, instead of the physical mass of receptors, ComRed is using the total intensity of objects in fluorescent pictures. The total intensity behaves analogous to the mass in the center of mass equations. The total intensity of an object is a product of the number of pixels or voxels and the mean intensity, which is analogous to the product density and volume for the total mass of an object. ComRed has two different metrics, based on center of mass calculations: (1) the center of mass of a given distribution, compared to a reference point, which might be inside or outside the cell. This is called reference metric. In this study, the center of mass of the nucleus was chosen as reference point. (2) The mean distance of every object of a given distribution to the center of mass of the whole distribution, defined as spread metric. The comparison of the center of mass to a reference point corresponds to a shift in 3D to the reference, while the mean distance corresponds to the spread of a receptor distribution. ComRed is also able to generate simulated data from experimental results. This might be useful to detect which magnitude of changes can be reliably detected in a specific set of data.

### 2.1. ComRed-Based Analysis Detects Changes in Experimental Data

Data simulations were used to show the strengths and limitations of ComRed. Simulated data were generated from experimental data by inverse transform sampling, i.e., a sampling method to generate a pool of random data points following an arbitrary distribution. Before the inverse transform sampling, the experimental data were modified by either shifting or spreading the function with a specific value *n*. A value of n=0 corresponds to no change, while higher *n* values correspond to a larger shift or spread in comparison to the original distribution (Section 4.9). After the inverse transform sampling, the differences between the raw experimental data distribution and the sampled distribution were measured with the cosine and Hellinger distances, as well as by using the percentage histogram overlap. All required parameters (x-, y-, and z-coordinates, volume, and mean intensity) were simulated from experimental data of BMPRIb distributions from 72 confocal microscopy images of HUVEC cells grown on circular micropatterned regions. First an inverse transform sampling mapping on the raw experimental data (n=0) was done. Examples of simulated distributions in comparison with experimental data are shown in Figure 2. Similarity metrics for the simulation in comparison with experimental data (for n=0) are available in Appendix A Figure A1. Subsequently, the simulated data with n-values ranging from 0 to 2 were analyzed and plotted with ComRed. Figure 3 and Figure 4 show a summary of the data simulation results, when compared to that of experimental data. In addition, experimental datasets were randomly split into two groups, which were compared to each other. The comparison of randomly split groups did not yield any significant differences in any of the test cases (Figure A2 and Figure A3). The reference metric showed significant differences for shifted data with n≥0.2, but no significant differences for data spread. Thus, the reference metric detects changes only on the distance of distributions to their reference point. The spread metric showed significant differences for data distribution with n≥0.2 For the shifted data significant differences (when applying a threshold of *p* < 0.05) for n=0, n=0.2, and n=1 were observed. So, while the spread metric behaves as expected for spreading distributions, it does not for the shifting of distributions. However, the distribution of mean distances of objects to their center of mass was much smaller for simulated data than for experimental data (Figure 4). These differences are not observed when comparing data split into two random groups (Figure A3). Therefore, the significant differences in shifted data might be an artifact due to the simulation behavior. In summary, ComReds reference metric is robustly detecting changes in the shift of a distribution, but not its spread. The spread metric on the other hand is robustly detecting differences in spread, but might also be sensitive to spread changes of the shift metric.

### 2.2. Cell Geometry Strongly Influences BMPR Subunit Distribution

After determining that the metrics used by ComRed are able to detect changes in localization and spread of receptors, ComRed was used to analyze distributions of BMPRIb and BMPRII in human umbilical vein endothelial cells (HUVECs). Three pairs of comparisons were done. BMP-treated cells were compared with control cells (no addition of BMP), so differences in distribution due to relocalization of receptors induced by the presence of the ligand could be analyzed. Circle- and crossbow-shaped cells were compared to analyze the differences in receptor distributions due to different cell geometries. BMPRIb and BMPRII distributions were compared to infer whether the two subunits might associate in a preformed complex (PFC) or in a BMP induced signaling complex (BISC), which both trigger different pathways. Should the distributions differ in shift or spread, the assembly of BISC might be preferred, as receptors are more likely to be in a dissociated state initially. A summary of the data produced by ComRed can be seen in Figure 5. The reference metric indicated no significant shift of distributions, except for BMPRIb and BMPRII in control crossbow cells (*p* = 0.025667). For the spread metric, three significant differences were observed when doing comparisons: in control circle cells, BMPRIb was more spread than BMPRII, and in both control and BMP-treated populations, circle and crossbow cells had a different distribution of BMPRIb. BMPRIb was more spread out than BMPRII in control cells ($p=1.067\cdot 10^{-3}$). However, this difference was no longer present upon addition of BMP, as in in the BMP induced case no significant difference between the receptor localizations could be found. This indicates that in HUVEC cells the BISC (and the corresponding Smad-independent pathway) might be preferred when cells have an unpolarized geometry, but not when in a polarized geometry, like the crossbow shape. Thus, the probability of BMP binding events, which lead to the activation of the Smad independent pathway, might be higher in control circle cells than in control crossbow cells. In BMP-treated cells, this discrepancy was no longer evident, as BMPRIb and BMPRII need to associate to trigger both signaling pathways. In control and BMP-induced cells, BMPRIb was more spread out in circle cells than in crossbow cells ($p=9.908\cdot 10^{-7}$ and $p=5.209\cdot 10^{-5}$ for control and BMP comparisons respectively). The localization of both receptors, BMPRIb and BMPRII appeared to be dependent on the same geometric constraints of the crossbow shape. If these changes were to depend only on the differences in cell geometries, also BMPRII would be more spread out in circular than in crossbow-shaped cells. Therefore, HUVEC cells might modulate the spread of BMPRIb based on polarization. Interestingly, there was no difference between the receptor distributions, when comparing control and BMP-treated cells. This suggests that a simultaneous internalization of the majority of receptors is unlikely.

Data generated with ComRed might be supplemented with external data. Here, we wanted to complete our data with knowledge whether the induced complex or the preformed complex is the dominant mode of interaction with BMP through image colocalization analysis. Colocalization was evaluated by using JaCoP [20], an ImageJ extension. Both signaling complexes should behave differently in a colocalization analysis. In the PFC the receptors are in close vicinity, because they bind to each other before BMP is binding to the complex. If a cell only uses the preformed complex, there should be no difference in the colocalization between the induced and noninduced state, because the complexes are already formed. In the BMP-induced signaling complex, both receptors are spatially separated and they only bind to each other when BMP is binding to BMPRI first. Both receptors might, however, be in close proximity without binding, so BISCs might appear like PFCs in the colocalization data. If however there is a significant difference between induced and noninduced states, there should be a preference for BISCs, as the receptor builds complexes only after BMP binding. Here, two different scores were used for validation, Pearson’s score (correlation) and Li’s image colocalization quotient (Li’s ICQ, colocalization). Four different comparison were done. Circle and crossbow cells were compared with and without BMP-treatment. In addition, scores for cells in presence or absence of BMP were compared for circle and crossbow shapes. The resulting data are presented in Figure 6. For the Pearson’s score no significant difference could be detected in any of the comparisons. For Li’s ICQ, two significant differences were detected. One between circle and crossbow cells in BMP- treated cells (*p* = 0.0079897) and one between control and BMP-treated circle cells (*p* = 0.0221327). The difference between control and BMP states agrees with the earlier results, where circle cells have a more spread out BMPRIb and the difference is no longer present after 15 min of BMP induction. The results from Li’s ICQ were however not reproducible with Pearson’s Score.

## 3. Discussion

In this study we observed differences in distributions between BMPRIb and BMPRII in circular, but not in crossbow cells. This difference was no longer present after induction with BMP2, indicating the formation of fully functional signaling complexes from BISCs. Benn et al. showed that the Smad-dependent response of HUVEC cells to BMP2 is very low, while there is a significant response in the non-Smad dependent pathway. Other cell types (human aortic endothelial cells and human pulmonary microvascular endothelial cells) react with both Smad-dependent and non-Smad dependent responses. It seems that the PFC for BMP-2 is strongly mediated by BMPRIa, which is expressed only weakly in HUVECs [12]. There is also a difference between circular and crossbow cells BMPRIb distribution. This might indicate that the preference for a specific pathway might not only be dependent on the type of cell and receptor, but also on the polarization of the cell. Although fully assembled BMPR complexes are known to internalize [29], their distributions are not compacting towards their center of mass. Therefore, we did not observe any coordinated internalization behavior in our experiments. It is unlikely that internalization requires a timeframe longer than the 15 min incubation we did in our study. It was reported that during the internalization of BMP (and as such the corresponding BMPRs), a peak could be reached already after 15 min [29]. Therefore, we propose that BMPRs might be recycled constantly. As receptors are internalized, other receptors are removed from an intracellular depot and are assembled in the cellular membrane. This replenishes the internalizing receptors and the spread with or without BMP induction does not change. A similar behavior can be seen in other membrane-bound proteins and receptors [30], like transforming growth factor β receptors (TGFβR) [31]. Mitchell et al. showed that TGFβRs are constantly internalized and relocalized to the surface, even in the absence of ligand [31]. TGFβ receptors and BMPRs are structurally related. As such, they form similar complexes of type I and type II receptors [32,33], as well as the internalization via clathrin-coated pit or caveolae [34]. Studies in *C. elegans* showed that deletion of key proteins in the recycling pathway lead to an accumulation of BMPR homologous proteins in vesicles with simultaneous depletion in the cell membrane [35]. These studies back up the claim of recycling BMPRs, but, to our knowledge, a definite analysis of the recycling behavior of BMPRs in mammalian cells was not done yet.

ComRed is a useful tool for generating hypothesis, rather than for testing hypothesis, on receptor distribution in cells. The program produces robust data based on data simulations compared to experimental data, represented by the analysis of datasets of at least 20 images. Thus, the use of ComRed might be challenging for the analysis of a low number of images and for experimental conditions where variability e.g., in cell shape and functions might strongly affect receptor distribution. As shown in Figure A3, the spread metric is strongly affected by differences in the spread of reference data. We applied ComRed for the analysis of confocal microscopy images, but our tool could be used with other microscopy-based methods, like FRET, FRAP, and fluorescence cross-spectroscopy, to gain a deeper understanding of cell receptors’ involvement in different processes.

## 4. Materials and Methods

### 4.1. Photo-Micropatterning

20 × 20 mm${}^{2}$ or 24 × 24 mm${}^{2}$ coverslips (Carl Roth, Karlsruhe, Germany) were placed in a sonication bath (Sonorex Super RK102H, Bandelin, Berlin, Germany) in ethanol for 30 min and dried. The surfaces were passivated with 0.1 mg/ml poly-L-lysine-polyethylene-glycol (PLL-PEG, Surface solutions, Dübendorf, Switzerland) in HEPES solution before photo-micropatterning using a UV-ozone cleaner (Model 342-220, Jelight Company Inc., Irvine, CA, USA). The surfaces were subsequently equilibrated in PBS and incubated with 0.0625 μg/cm${}^{2}$ human cellular fibronectin (Sigma Aldrich, St. Louis, MO, USA) in PBS for one hour before washing with PBS. For the experiments a circle- and crossbow-pattern was used, where every spot has a diameter of 50 µm, to ensure only one cell spreads on every micropattern.

### 4.2. Cell Culture

Human umbilical vein endothelial cells (HUVECs, pooled donors cryopreserved, catalog number C-12203; PromoCell, Heidelberg, Germany) were cultured in T25 flasks (Greiner Bio One, Frickenhausen, Germany) using Endothelial cell basal medium 1/2 (PromoCell, Heidelberg, Germany). They were passaged using Accutase (Thermo Fisher Scientific, Waltham, MA, USA) for detachment. Cells were seeded on 20 × 20 mm${}^{2}$ or 24 × 24 mm${}^{2}$ coverslips (Carl Roth, Karlsruhe, Germany) in 6-well plates (Thermo Fisher Scientific, Waltham, MA, USA). Cells were fixed using 4% (*w*/*v*) paraformaldehyde (Sigma Aldrich, St. Louis, MO, USA) in PBS at room temperature for 15 min.

### 4.3. Immunofluorescence Staining

Paraformaldehyde fixed cells on coverslips were permeabilized with 0.1% (*v*/*v*) Triton-X-100 (Sigma Aldrich, St. Louis, MO, USA) in PBS (*v*/*v*) solution for 10 min. Subsequently, the surfaces were blocked with 2% BSA (Carl Roth, Karlsruhe, Germany) in PBS solution. The surfaces were incubated with primary and then secondary antibody solution for one hour. Actin was stained using Phalloidin-TRITC (P1951, Sigma Aldrich, St. Louis, MO, USA) at 5 µg/mL, BMPRIb was stained using anti-BMPRIb mouse mIgG (MAB5051, R&D Systems, Minneapolis, MN, USA) at 5 µg/mL, p-Smad was stained using anti p-Smad1/5 (Ser463/465) rabbit IgG (9516, Cell signaling, Danvers, MA, USA) at 1 µg/mL, BMPRII was stained using anti-BMPRII rabbit pIgG (AP11864PU-N, Acris Antibodies, Herford, Germany) at 2.5 µg/mL. As secondary antibodies we used Alexa647 goat anti-rabbit IgG (A-21244, Thermo Fisher Scientific, Waltham, MA, USA), and Alexa488 goat anti-mouse IgG (A-11029, Thermo Fisher Scientific, Waltham, MA, USA) at 10 µg/mL. After primary and secondary antibody staining, the coverslips were washed in 2% BSA in PBS. Stained coverslips were embedded in mowiol (Carl Roth, Karlsruhe, Germany) supplemented with DABCO (Sigma Aldrich, St. Louis, MO, USA) and 1 µg/ml DAPI (Thermo Fisher Scientific, Waltham, MA, USA).

### 4.4. Microscopy

Confocal images were acquired using a Zeiss LSM880 microscope (Zeiss, Germany). z-stack confocal images of single cells were obtained using a 63× oil objective. Only cells with the desired shape (circle or crossbow) were imaged. Micropatterned regions with more than one attached cell were excluded.

### 4.5. Seeding Human Umbilical Vein Endothelial Cells on Micropatterns and Induction with Bone Morphogenetic Protein 2 (BMP-2)

Human umbilical vein endothelial cells (HUVECs) were cultured according to the Section 4.2. HUVECs were seeded on circle or crossbow micropatterned surfaces (Section 4.1). Cells were allowed to spread on the micropattern for 45 min. Afterwards, either bone morphogenetic protein 2 (BMP-2) derived from Chinese hamster ovary cells (CHO cells; R & D Systems, Minneapolis, MN, USA) to a final concentration of 20 nM or a drop without BMP was added (control condition) and the cells were incubated for another 15 min. The cells were fixed with paraformaldehyde and subsequently immunostained (Section 4.2 and Section 4.3).

### 4.6. Image Processing and Analysis

Before using the center of mass calculations, confocal microscopy pictures were processed with two different programs: ImageJs Fiji distribution (Version 1.51) [36] and ilastik (Version 1.2) [26,27]. Fiji was used for image preprocessing, where pictures were converted to grayscale images and contrast was adjusted. Ilastik, a machine learning framework for biomedical imaging, was used for object detection of receptor accumulations. The 3D positions, volumes and mean intensities for every receptor accumulation were determined. Colocalization was calculated with the JaCoP imageJ extension [20]. For the colocalization the Pearson’s Score and Li’s ICQ were used. For Li’s Score first calculates $\left(A_{i}-a\right)\left(B_{i}-b\right)$, where $A_{i}$ and $B_{i}$ are pixel intensities of the same pixel in two channels and *a* and *b* are the means of intensities in one channel in the whole picture. Then, the quotient of the positive values to all values is calculated. In the end, also 0.5 is subtracted from the result, so the ICQ ranges between −0.5 and 0.5 [19]. Pearson’s Score is a correlation score, which is defined as $P=\frac{\sum \left(A_{i}-a\right)\left(B_{i}-b\right)}{\sqrt{\sum {\left(A_{i}-a\right)}^{2}}\cdot \sqrt{\sum {\left(B_{i}-b\right)}^{2}}}$, ranges from 0 to 1.

### 4.7. Center of Mass Calculations

The general formula for center of mass calculations is given as: $r=\frac{\sum m_{i}\cdot r_{i}}{\sum m_{i}}$. Where $m_{i}$ is the mass of every object and $r_{i}$ is its position in one spatial direction and *r* is the position of the center of mass. For pictures the real mass is substituted by the virtual mass given as: $v_{i}={\bar{I}}_{i,g}\cdot n_{i}$, where ${\bar{I}}_{i,g}$ is the mean intensity of the object normalized over the mean intensity of the picture (${\bar{I}}_{i,g}=\frac{I_{i}}{I_{g}}$ and $n_{i}$ is the object volume in voxels, in our case 0.0037 µm^3^). The virtual mass $v_{i}={\bar{I}}_{i,g}\cdot n_{i}$ is analogous to the real world mass, which is given by density times volume. Inserted into the original center of mass formula $I_{g}$ cancels out, leaving $r=\frac{\sum I_{i}\cdot n_{i}\cdot r_{i}}{\sum I_{i}\cdot n_{i}}$. Pictures might differ in size (physical dimensions, and/or pixel volume, and/or pixel number), and cell orientation and position. Therefore object coordinates or center of mass coordinates need to be compared to a frame of reference. $d=r_{o}-r_{r}$ where *d* is the difference between an object and its reference, $r_{r}$ is the reference position and $r_{o}$ is the object position. After the positions for all spatial directions were calculated the arithmetic mean might be calculated $\bar{d}=\sqrt{{\left(d_{x}\right)}^{2}+{\left(d_{y}\right)}^{2}+{\left(d_{z}\right)}^{2}}$. In the end, the difference in pixels might can be converted into a length scale of choice (here micrometer). In this study, we used two metrics-a reference metric and a spread metric. In the reference metric, the object (as indicated by $r_{o}$ is the center of mass of a receptor distribution, while the reference ($r_{r}$) is the center of mass of the cells nucleus). In the spread metric, the object is every receptor accumulation detected (so every spot on the picture would be one object), and the reference is the center of mass of all found accumulations of the same receptor type. Then, the mean of the spread values was calculated for every cell. The current version of ComRed is available through https://github.com/hendrikboog/comred (last accessed: 13 October 2021).

### 4.8. Statistics

Statistics were done with the scipy package for python using a two-sided Mann–Whitney-U test.

### 4.9. Data Simulation

To test the boundaries of the data analysis tool, data was simulated by mimicking experimental data. This was done with inverse transform sampling. With this method one might generate random numbers from an arbitrary probability distribution. In this case, the probability distribution is the distribution of the position of single receptor accumulations, their volumes, or mean intensities. To get a random distribution the equation $F(F^{-1}\left(y\right))=y$ needs to be solved for $F^{-1}\left(y\right)$, where F(x) is the non-inverse distribution and $F^{-1}\left(y\right)$ is the inverse transform of the distribution. The inverse transform of a probability distribution can now be fed with random variables from a uniform distribution between 0 and 1 to get random numbers that fit the distribution F(x). As a subsequent step the input data might be altered, so that different cases of experimental versus model data can be examined. Here, we focused on displacement of the function and a change in spread (effectively “flattening”) of the function. These two cases represent a change in the center of mass and widely distributed receptors, respectively. For the first, every experimental input data point was altered with $x_{new}=x_{exp}+n\cdot \bar{x}$, where $x_{new}$ is the altered data point, $x_{exp}$ is the read in datapoint from experimental data, *n* is an arbitrary real number and $\bar{x}$ is the mean of all experimental data points $x_{i}$. For the change in spread, the following formula was used: $x_{new}=x_{exp}+\frac{n\cdot \bar{x}\cdot \left(x_{rnd}-\bar{x}\right)}{b}$, where $x_{rnd}$ is a random number drawn from an inverse transform sampling of the experimental data and *b* is the longest distance between the mean of data and the outermost data point. Afterwards simulated data were produced by inverse transform sampling of the altered data. Both receptor distributions and reference points were simulated. Reference points were always simulated with *n* = 0, while the receptors were simulated with n-values from 0 to 2. Discrepancies between simulated and experimental data were analyzed with the cosine distance, a percentual histogram overlap, and the Hellinger distance. The cosine distance is the cosine of the angle between two vectors of data. The distance ranges between 0 and 1, where 0 is no change. The cosine distance is independent of the vectors length. The histogram overlap is the normalized integral between both histogram functions. A value of 1 is a perfect overlap and a value of 0 means no overlap. The Hellinger distance for discrete distributions is defined as $H\left(P,Q\right)=\frac{1}{\sqrt{2}}\sqrt{\sum_{i=1}^{k}{\left(\sqrt{p_{i}}-\sqrt{q_{i}}\right)}^{2}}$, the larger the Hellinger distance, the more distinct the two distributions are.

## Figures and Tables

**Figure 1 jimaging-07-00219-f001:**
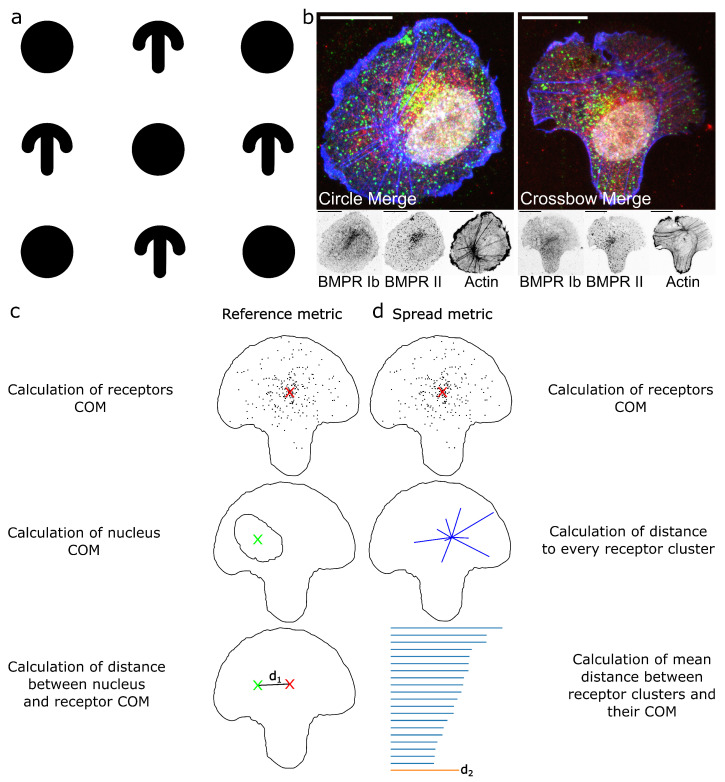
Outline of experiments and mode of operation of ComRed. (**a**): Scheme of circle and crossbow pattern. White background is passivated with PLL-PEG, while black regions are micropatterned using a photomask in a UV-ozone cleaner and coated with fibronectin (Section 4.1). Circle diameter is 50 µm. (**b**): examples of confocal microscopy images of HUVECs seeded on a circle (left) and crossbow micropattern (right). Green: BMPRIb, Red: BMPRII, Blue: actin, White: nucleus. Scale bars in all confocal images are 20 µm. (**c**,**d**): steps for the calculation of reference and spread metrics. For both metrics, first the center of mass of a receptor distribution is calculated. (**c**): For the reference metric, a reference point is taken, e.g., the center of mass of the nucleus. Afterwards, the distance between the two center of mass points is calculated. (**d**): For the spread metric, after determination of the center of mass of the receptor distribution, the distance to each single accumulation is calculated and the mean of these distances is taken.

**Figure 2 jimaging-07-00219-f002:**
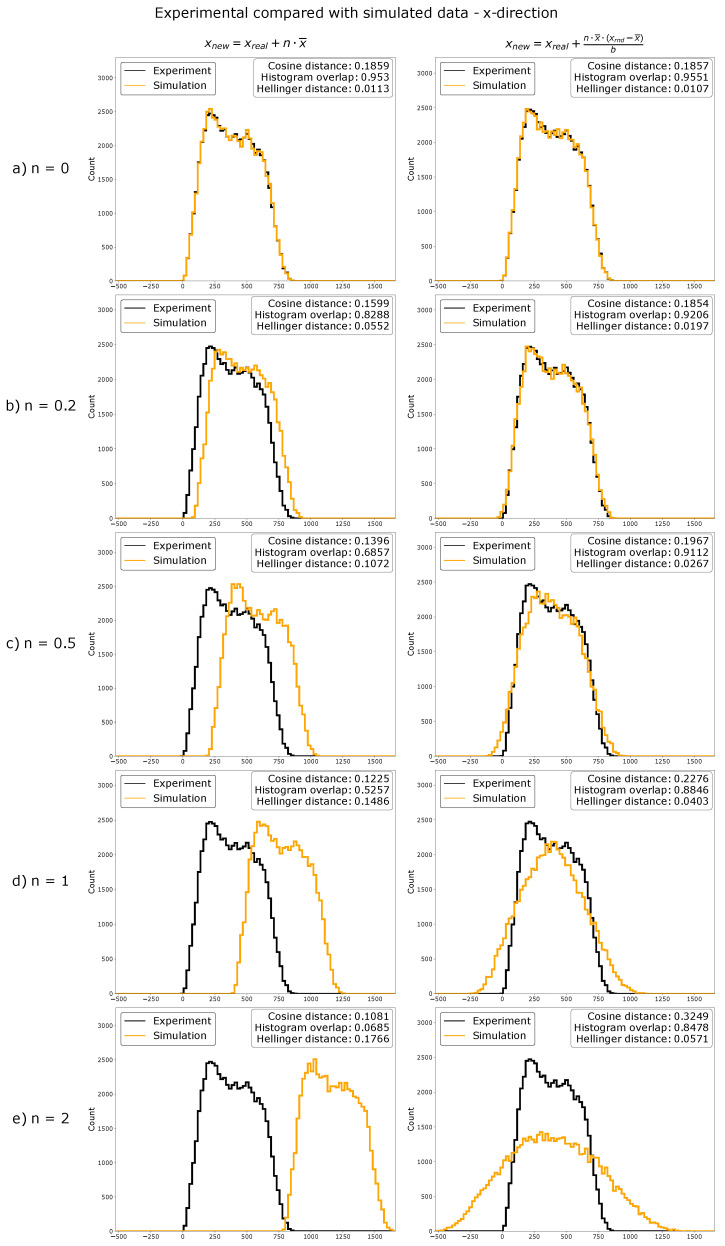
Overlaid histograms of experimental data (black) and simulated data (orange) transformed with factor *n*. Experimental data were used to generate simulated data, as explained in Section 4.9. Different magnitudes of changes (as indicated with their *n*-value) were tested. Cosine distance and Hellinger distance, as well as histogram overlap were calculated. Here, data can be effectively simulated by inverse transform sampling. Distributions can be effectively shifted or spread by using functions $x_{new}=x_{exp}+n\cdot \bar{x}$ (for shifting of distributions) or $x_{new}=x_{exp}+\frac{n\cdot \bar{x}\cdot \left(x_{rnd}-\bar{x}\right)}{b}$ (for spreading of distributions) before inverse transform sampling.

**Figure 3 jimaging-07-00219-f003:**
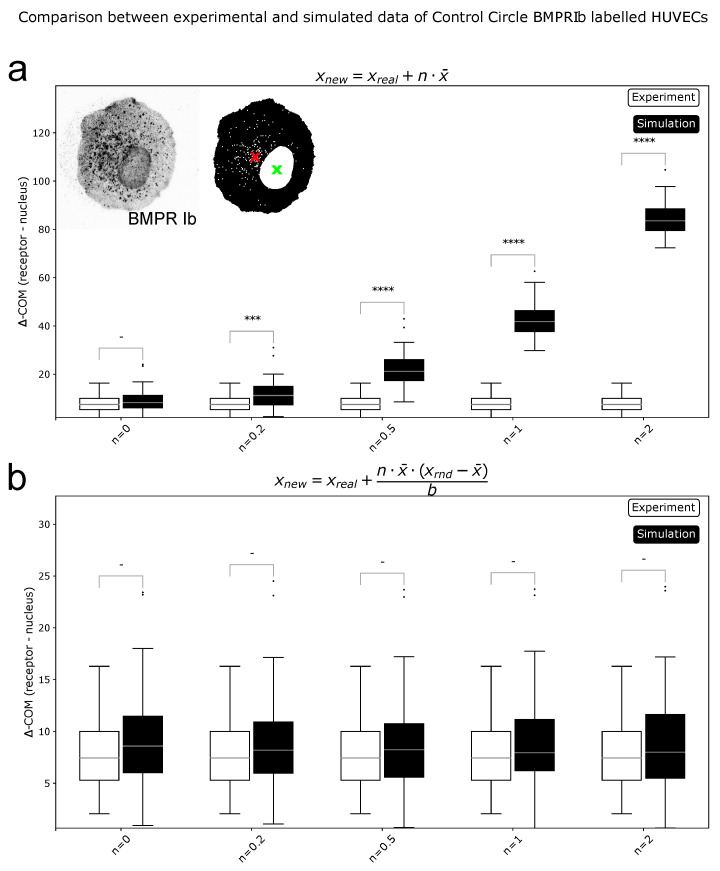
Comparison of simulated and experimental data for reference metric. (**a**): Simulations of a shift in distributions ($x_{new}=x_{real}+n\cdot \bar{x}$). All tested values for *n* > 0 showed significant differences. (**b**): Simulations of a spread in distributions ($x_{new}=x_{real}+\frac{n\cdot \bar{x}\cdot \left(x_{rnd}-\bar{x}\right)}{b}$). Here, for all values of *n* no significant differences were found. Reference metric is sensitive to shifts in COM position, but not to differences in spread of distributions. *** = *p* < 0.001, **** = *p* < 0.0001, - = not significant.

**Figure 4 jimaging-07-00219-f004:**
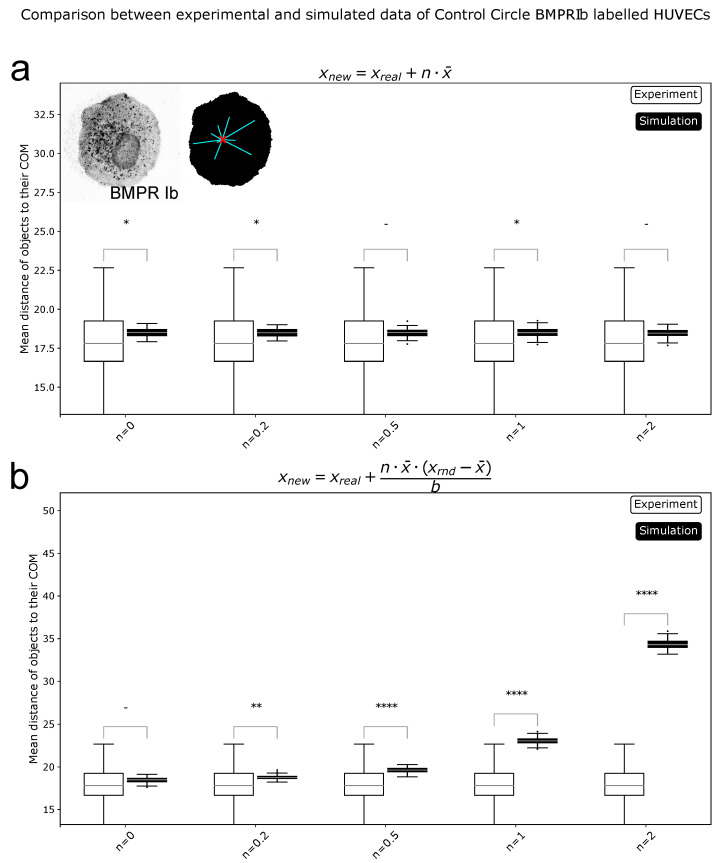
Comparison of simulated and experimental data for spread metric. (**a**): Simulations of a shift in distributions ($x_{new}=x_{real}+n\cdot \bar{x}$). (**b**): Simulations of a spread in distributions ($x_{new}=x_{real}+\frac{n\cdot \bar{x}\cdot \left(x_{rnd}-\bar{x}\right)}{b}$). Here, for all values of *n* > 0 significant differences were found. Spread metric is sensitive to the spreading of distributions. * = *p* < 0.05, ** = *p* < 0.01, **** = *p* < 0.0001, - = not significant.

**Figure 5 jimaging-07-00219-f005:**
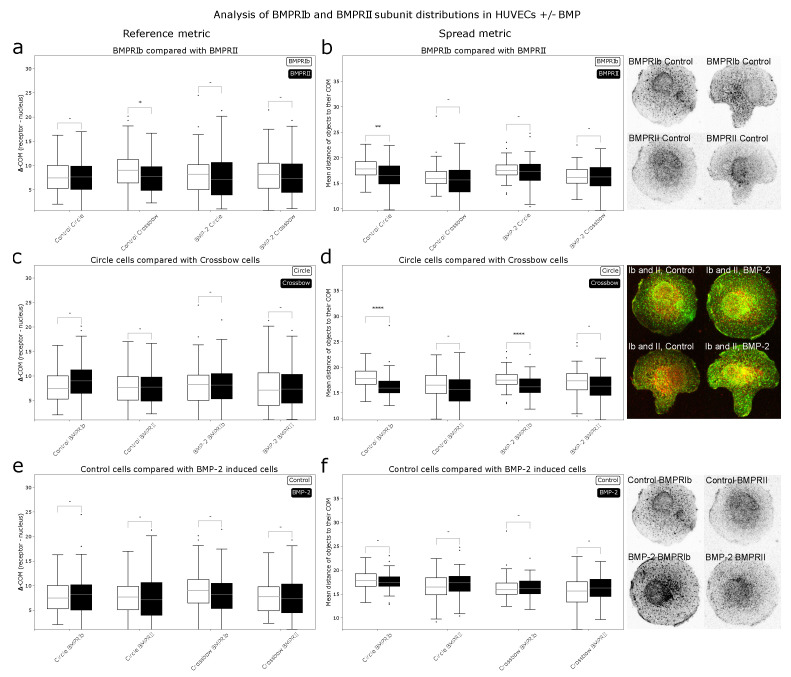
Comparison of receptor distributions of BMPR subunits in HUVECs. (**a**,**c**,**e**): reference metric data, (**b**,**d**,**f**): spread metric data. Confocal images of cells adhering to micropatterns: Green: BMPRIb, red: BMPRII. (**a**,**b**): distributions of BMPRIb compared to distributions of BMPRII. Differences in shift could be observed in crossbow cells under control conditions (p≈0.026). Also, differences could be observed between BMPRIb and BMPRII ($p\approx 10^{-3}$) in circular cells without BMP, indicating a preference for BISC. (**c**,**d**): Comparison of circle and crossbow cells. No significant differences in shift could be found. BMPRIb is more spread out in circle than in crossbow cells, both under control and BMP conditions ($p\approx 10^{-6}$ and $p\approx 5\cdot 10^{-5}$ respectively). (**e**,**f**): control cells compared with cells that were induced with BMP-2. Control and BMP-2 treated cells do not show a significant difference in receptor distribution; this suggests, that majority of BMPRs were not internalizing simultaneously after addition of BMP. * = *p* < 0.05, ** = *p* < 0.01, **** = *p* < 0.0001, - = not significant.

**Figure 6 jimaging-07-00219-f006:**
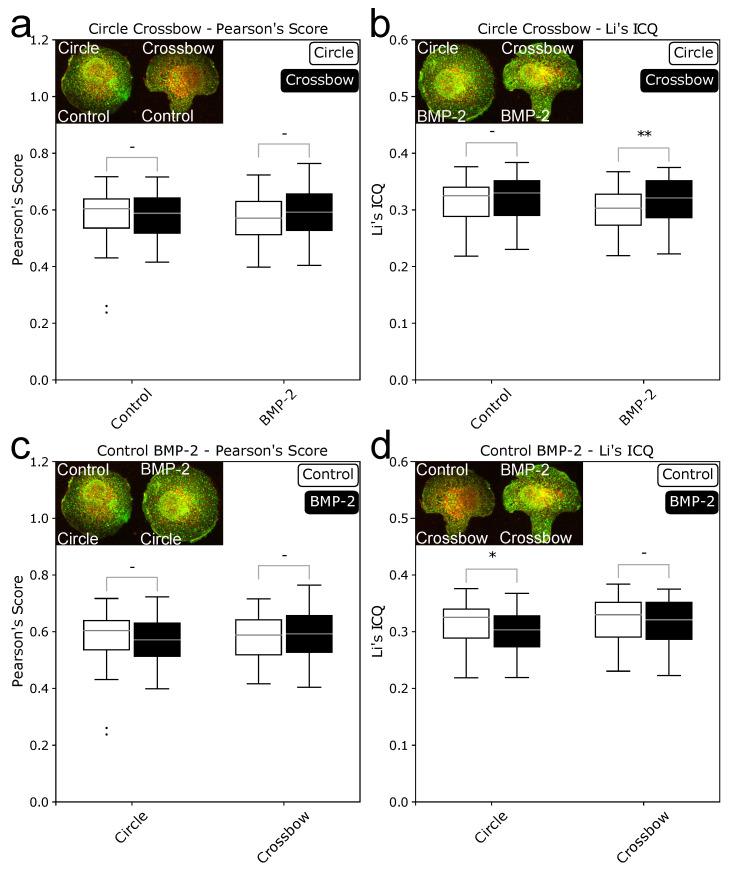
Boxplots of colocalization values of BMPRIb and BMPRII in HUVECs. (**a**,**b**): colocalization values of circle and crossbow cells +/− BMP-2, using Pearson’s score (**a**) and Li’s ICQ (**b**). c and d: colocalization values of control and BMP-2 induced cells in circle and crossbow cells, using Pearson’s score (**c**) and Li’s ICQ (**d**). Confocal images of HUVECs on circle- and crossbow-patterns were analyzed with the JaCoP imageJ extension [20]. Changes in colocalization between induced an noninduced cells indicated that the BMP-induced signaling complex is preferably used. No change in colocalization suggested that preformed complex is predominantly used. * = *p* < 0.05, ** = *p* < 0.01, - = not significant.

## Data Availability

ComRed is available at https://github.com/hendrikboog/comred (last accessed: 13 October 2021).

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
