# Peer review of "Single Cell Center of Mass for the Analysis of BMP Receptor Heterodimers Distributions"

_2313-433X, 2021, doi:10.3390/jimaging7110219_

Round 1
Reviewer 1 Report
In this manuscript the authors investigated the distribution of BMPRIb and BMPRII on HUVEC cells plated on circular and on crossbow micron-patterns, before and after stimulation with soluble BMP2. The authors adopted a script called ComRED, developed to simulate first and analyze afterwards any differences in the localization and spread of receptors (or any other labeled protein for future application). The authors initially proved that the script is well adapted to detect changes in the distribution of the receptors. The ComRED analysis showed that BMPRIb is more spread out than BMPRII in circle patterns but not in crossbow and that the distribution of BMPRIb is different between the two different patterns (more spread out in circular than on crossbow shape). The analysis of the co-localization suggested that HUVEC plated on circular shape present preferentially a BMP induced signaling complex. This shows that not only the type of the cell but also their polarization induce one or another BMP signaling cascade.
I believe this work is convincing and the base for future studies using labeled endogenous receptors. The introductions and the discussion are well written and complete. My main concerns the explanation of the results that is not easy to follow. To improve the understanding of the results it is important to better present the figures. Their description is indeed not following the chronological order. Moreover figure 3 and 4 are not cited. I would suggest to place then in the attached images if no needed to explain the main message. The figures 5 and 6 report the same images (at least they look like the same but no explanation permit to understand if they are different) on the right, I would remove this repetition. Please also control the figure captions.
Author Response
We would like to thank the reviewer for the positive comments on our manuscript and greatly appreciate his suggestions to improve the explanation of our results.
- We have revised the initial introductory part of the result section relative to Figure 1
- We have corrected the citation of Figures, which are now indicated and described them in the correct chronological order
- We removed the repetition of the images and combined Figures 5 and 6 into one single figure
- The figure captions have been revised to avoid repetitions and make the explanation of the data more clear.
Reviewer 2 Report
The description of Figure 2 is a bit misleading, I wouldn't call these plots boxplots, rather just histograms. Please check for other typos or similar errors.
Figure 4 is not referenced in the text. Please do so with additional explanations.
Author Response
We thank the reviewer for pointing out the misleading description of Figure 2 and the missing information about Figure 4 in the text.
- We have improved the description in the text for all the figures, following their chronological order of appearance
- We rephrased the description of Figure 2 to improve clarity and understanding of the data presented and apologise for the mistake in indicating the histograms as box plots.
- We revised the text for the description of Figures 3 and 4, which were erroneously referenced.
- We corrected the entire manuscript for typos and grammar errors, in all its sections.
Reviewer 3 Report
The authors have developed a technique called ComRed to analyze the distribution of receptors proteins in the cells. They authors demonstrated that the receptor distribution (such as BMPR) is dependent on cell polarization. The authors investigated a novel technique, which can be used to study the local regulation of different signaling receptor molecule. However, some minor points for improvement have to be considered.
1. The author has described the advantages of ComRed in the discussion part. It would be great, if they can also include some weaknesses of the ComRed technique in the discussion.
2. Since there are several HUVECs cells are available in the Promocell, please include the catalogue number for the HUVECs cells along with the company name in the methodology section.
3. Figure 4 is not reference in the manuscript.
4. Please mention the p values in all the figure legend (e.g * p < 0.05, ** p< 0.005).
Author Response
We would like to thank the reviewer for the positive comments and are grateful for the comments that helped us improving the quality of our paper. Please find our responses (text in blue) below each point raised by the reviewer:
1. The author has described the advantages of ComRed in the discussion part. It would be great, if they can also include some weaknesses of the ComRed technique in the discussion.
We revised the last paragraph in the discussion section, page 9, line 219-227, elaborating on the weaknesses and limitations of ComRed.
2. Since there are several HUVECs cells are available in the Promocell, please include the catalogue number for the HUVECs cells along with the company name in the methodology section.
We thank the reviewer for pointing out that the information about the HUVECs was missing, and we inserted the details on the company name and catalogue number in the Materials and Methods section, page 12, line 239-240.
3. Figure 4 is not reference in the manuscript.
Figure 4 is now mentioned in the main text on page 3, line 120, and page 5, line 131
4. Please mention the p values in all the figure legend (e.g * p < 0.05, ** p< 0.005).
We inserted the information about the p values in all the figure legends.